No evidence for asymmetric sperm deposition in a species with asymmetric male genitalia

van Gammeren Sanne 1
Lang Michael 2 3
Rücklin Martin 1
Schilthuizen Menno menno.schilthuizen@naturalis.nl 1 4
1 Naturalis Biodiversity Center , Leiden , The Netherlands
2 Université Paris Cité, CNRS - Institut Jacques Monod , Paris , France
3 Institut Diversité, Ecologie et Evolution du Vivant (IDEEV), Laboratoire Évolution, Génomes, Comportement et Écologie, CNRS, IRD, Université Paris-Saclay , Gif-sur-Yvette , France
4 Institute for Biology, Leiden University , Leiden , Netherlands
Rogers Lesley
Electronic publication date: 2022 Nov 24
Publication date: 2022
Volume: 10
Electronic Location ID: e14225
Received 2022 Apr 12; Accepted 2022 Sep 21
Copyright: ©2022 van Gammeren et al.
Copyright year: 2022
Copyright holder: van Gammeren et al.
License: This is an open access article distributed under the terms of the Creative Commons Attribution License, which permits unrestricted use, distribution, reproduction and adaptation in any medium and for any purpose provided that it is properly attributed. For attribution, the original author(s), title, publication source (PeerJ) and either DOI or URL of the article must be cited.
License URL: https://creativecommons.org/licenses/by/4.0/

Keywords: Asymmetric genitalia, Drosophila pachea, Sperm transfer, Copulatory complex, X-ray computed tomography

Funding: Naturalis Biodiversity Center CNRS Agence Nationale de la recherche ANR-20-CE13-0006 Paul Scherrer Institute This work was supported by the Naturalis Biodiversity Center and the CNRS. A part of this work was also supported by a grant of the Agence Nationale de la recherche [ANR-20-CE13-0006] given to Michael Lang. Beamtime at the Paul Scherrer Institute was granted to Martin Rücklin and Menno Schilthuizen. The funders had no role in study design, data collection and analysis, decision to publish, or preparation of the manuscript.

==============================
Background

Asymmetric genitalia have repeatedly evolved in animals, yet the underlying causes for their evolution are mostly unknown. The fruit fly Drosophila pachea has asymmetric external genitalia and an asymmetric phallus with a right-sided phallotrema (opening for sperm release). The complex of female and male genitalia is asymmetrically twisted during copulation and males adopt a right-sided copulation posture on top of the female. We wished to investigate if asymmetric male genital morphology and a twisted gentitalia complex may be associated with differential allocation of sperm into female sperm storage organs.

Methods

We examined the internal complex of female and male reproductive organs by micro-computed tomography and synchrotron X-ray tomography before, during and after copulation. In addition, we monitored sperm aggregation states and timing of sperm transfer during copulation by premature interruption of copulation at different time-points.

Results

The asymmetric phallus is located at the most caudal end of the female abdomen during copulation. The female reproductive tract, in particular the oviduct, re-arranges during copulation. It is narrow in virgin females and forms a broad vesicle at 20 min after the start of copulation. Sperm transfer into female sperm storage organs (spermathecae) was only in a minority of examined copulation trials (13/64). Also, we found that sperm was mainly transferred early, at 2–4 min after the start of copulation. We did not detect a particular pattern of sperm allocation in the left or right spermathecae. Sperm adopted a granular or filamentous aggregation state in the female uterus and spermathecae, respectively.

Discussion

No evidence for asymmetric sperm deposition was identified that could be associated with asymmetric genital morphology or twisted complexing of genitalia. Male genital asymmetry may potentially have evolved as a consequence of a complex internal alignment of reproductive organs during copulation in order to optimize low sperm transfer rates.

Introduction

Animal genitalia are often remarkably complex and reveal a high degree of variation between species (Eberhard, 1985). Sexual selection has been proposed as the chief driver for both the rapid diversification and the great morphological complexity of genital structures through sexual conflict between females and males at different levels of reproduction, including access to copulation, sperm competition inside the female, and control over post-mating sperm storage and usage for fertilization (Thornhill, 1983; Eberhard, 1985; Arnqvist, 1998; Birkhead & Pizzari, 2002; Chapman et al., 2003). In general, genitalia enable ejaculate transfer during copulation from the male into the female and mediate inter-sexual communication during copulation (Eberhard, 1985; Eberhard, 1994).

Asymmetric genitalia are observed in many species and must have recurrently evolved from symmetric ancestors (Huber, Sinclair & Schmitt, 2007; Huber, 2010; Schilthuizen, 2013). They have been associated with a specific interlocking of female and male genitalia (Kamimura, Yang & Lee, 2019; Holwell et al., 2015; Rhebergen et al., 2016; Richmond, Park & Henry, 2016) and with lateralized courtship and copulation behavior (Rhebergen et al., 2016; Orbach et al., 2020; Torres-Dowdall et al., 2020). However, the ultimate evolutionary cause for these transitions from symmetry to asymmetry are not fully understood (Schilthuizen, 2013). To help solve this puzzle, detailed analyses of the function and the internal configuration of asymmetric genitalia during copulation are helpful, as already evaluated in a few cases. In anseriform birds (waterfowl), the female vagina is coiled clockwise and has dead-end sacs that potentially function to avoid insemination via the male phallus, which is coiled in counter-clockwise direction (Brennan et al., 2007). Similarly, harbor porpoise Phocoena phocoena (L.) reveal strikingly asymmetric female and male genitalia (Orbach et al., 2020) and males attempt copulation by approaching a female exclusively on her left side, potentially to overcome female insemination or fertilization barriers. Lateralized mating behavior has also been shown to be associated with genital asymmetry in one-sided livebearer fish Jenynsia lineata (Jenyns) (Torres-Dowdall et al., 2020). The tip of the male gonopodium is either bent to the left or to the right side and the direction of this asymmetry was correlated with sidedness of mating attempts. Also, complementary antisymmetric genitalia have been suggested to increase reproductive success in snails of the subgenus Amphidromus s. str. The genitalia of these hermaphrodite species are coiled and favor copulation of interchiral pairs that better match for sperm transfer and that circumvent sperm digestion (Schilthuizen et al., 2007; Schilthuizen & Looijestijn, 2009). In insects, the evolution of asymmetric genitalia was proposed to have evolved in response to changes in mating position, potentially affecting the coupling efficiency of female and male genitalia during copulation (Huber, 2010). Changes in mating positions could evolve through sexual conflict, for the male to gain control over copulation, or through the female to increase the difficulty of insemination and to increase the potential to reject male gametes. This may impose selection pressure on genitalia to become asymmetric to modify genital contacts. However, the opposite is also possible, that asymmetric genitalia may provoke adaptive changes in mating position. In addition, no experimental data exist to test this hypothesis and generally little is known about the internal alignments of the intermittent phallus inside the female reproductive tract. Such studies have, however, become feasible in recent years due to the advances in micro-computed tomography (Mattei et al., 2015; Woller & Song, 2017; Dougherty & Simmons, 2017; Gutiérrez et al., 2018).

Males of Drosophila pachea Patterson & Wheeler have strikingly asymmetric genitalia. They possess a pair of external genital lobes that are attached to the lateral plates of the male genital arch (= epandrium: Rice et al., 2019). The left lobe being approximately 1.5 times longer than the right lobe (Pitnick & Heed, 1994; Lang & Orgogozo, 2012). In addition, the phallus is asymmetrically bent, harbors ventrally asymmetric spurs and the male phallotrema (opening for sperm release) is positioned at the right dorsal tip of the phallus (Acurio et al., 2019). Apart from this asymmetric genital morphology, D. pachea also mates in a right-sided copulation posture, with the male antero-posterior midline being shifted about 6−8° to the right side of the female midline (Lang & Orgogozo, 2012; Rhebergen et al., 2016; Acurio et al., 2019). This one-sided mating posture is associated with asymmetric genitalia coupling and an asymmetric twist of the female ovipositor (Rhebergen et al., 2016), potentially resulting in a specific positioning of the male phallotrema inside the female reproductive tract. Right-sided mating and male genital asymmetry have co-evolved in D. pachea and closely related species, but so far it has not been determined whether right-sided mating behavior may have provoked the evolution of morphological asymmetry or vice versa (Acurio et al., 2019). Within the genus Drosophila, D. pachea belongs to the nannoptera species group, which consists of four described species: D. nannoptera Wheeler has fully symmetric genitalia but copulates in a right-sided mating posture, D. acanthoptera Wheeler has an asymmetric phallus but mates in a symmetric posture, and D. wassermani Pitnick & Heed has asymmetric anal plates and an unknown copulation posture. Thus, right-sided mating and male genital asymmetry appears to have evolved repeatedly in this species group.

Sperm storage and usage for fertilization has been associated with sexual conflict and the evolution of sexual characters with respect to sperm selection from multiple copulations, sperm viability and temporal unlocking of copulation and fertilization (Orr & Zuk, 2012; Edward, Stockley & Hosken, 2015). Drosophila species store sperm after copulation in two types of sperm storage organs, the tubular seminal receptacle and the paired spherical spermathecae (Pitnick, Markow & Spicer, 1999). Both types of organs are connected to the anterior end of the female uterus. In particular, D. pachea, D. acanthoptera and D. wassermani were reported to store sperm exclusively in the spermathecae (Pitnick, Markow & Spicer, 1999). Sperm length also varies enormously among Drosophila species and range from 0.3 mm in D. persimilis Dobzhansky & Epling to 58 mm in D. bifurca Patterson & Wheeler (Pitnick, Markow & Spicer, 1995). In particular, D. pachea and its sister species D. nannoptera also produce giant sperm with lengths of 16.5 and 15.7 mm, respectively (Pitnick & Markow, 1994). Migration capacities of such long sperm require further investigation and it is unknown how giant sperm is organized post copulation inside female reproductive tract.

It is known that in certain Diptera females preferentially store sperm in specific spermathecae for high-quality males (Otronen, 1998; Hellriegel & Bernasconi, 2000). It is therefore conceivable that the asymmetric male genitalia in D. pachea may have evolved to enhance the chances that sperm would be deposited in or near such preferential sperm storage organs. The aim of our study was to investigate if male genital asymmetry results in an asymmetric internal configuration of the copulatory complex in D. pachea which may allow sperm to be directed specifically towards one of the female spermathecae. We carried out micro-computed tomography (microCT) and synchrotron-based X-ray computed tomography of snap-frozen D. pachea mating couples before and during copulation to investigate the positioning of reproductive organs as well as ejaculates during copulation. Furthermore, we investigated when sperm transfer occurs during copulation and what aggregation states ejaculate mass assumes during and after transport into the spermathecae.

Materials & Methods

Drosophila pachea maintenance and virgin collection

Drosophila pachea stock 15090-1698.02 from the San Diego Drosophila Species Stock Center (now The National Drosophila Species Stock Center, College of Agriculture and Life Science, Cornell University, USA) was maintained in 25 × 95 mm plastic vials containing 10 mL food medium (60 g/L brewer’s yeast, 66.6 g/L cornmeal, 8.6 g/L agar, 5 g/L methyl-4-hydroxybenzoate and 2.5% v/v ethanol). In addition, 40 µL of 5 mg/mL 7-dehydrocholesterol (dissolved in ethanol) was mixed into the food medium with a spatula (standard D. pachea food). Flies were transferred to fresh vials every 2 to 4 days. In order to isolate virgin individuals, adult flies at 0–3 day after emerging from the puparia were anaesthetised on a CO2-pad (INJECT+MATIC Sleeper) under a stereo-microscope Stemi 2000 (Zeiss), separated according to sex and maintained in groups of 20–30 female or male individuals until they reached sexual maturity, about 14 days for males and four days for females (Pitnick, 1993).

micro-CT scanning

Sexually mature virgin couples of D. pachea were introduced each into a 2-mL reaction tube (Eppendorf) that was roughened on the inside. Flies were observed at room temperature for at most 1 h or until copulation had ended. As soon as a couple started to copulate, time was recorded and the couple was snap-frozen by submerging the Eppendorf tube for 20 s in liquid nitrogen after one of the following intervals: 2 min, 8 min, 15 min, 20 min, and 120 min after the start of copulation. Tubes were filled with cold (−20 °C) absolute ethanol and were stored at −20 °C for about a week. Finally, the absolute ethanol was replaced with 80% ethanol and samples (mating complexes) were stored at room temperature. A total of 40 samples were prepared for micro-computed tomography scanning (micro-CT) by critical point-drying, including 25 mating complexes, nine single females two hours after copulation start (ACS) and six virgin females (Table 1). First, the complexes were fixed by successive 20-min incubations in 90% ethanol, 96% ethanol, and twice in absolute ethanol. Samples were then dried in the Automated Critical Point Dryer EM CPD300 (Leica) (program 1, parameters: ‘CO2 IN’ speed slow, delay 120 s; ‘Exchange’ speed 5, cycles 12; ‘gas OUT’ heat slow, speed slow). Samples were subsequently mounted on an aluminium stub with the male’s dorsal side (mating complexes) or the female dorsum (for single female specimen) glued to the stub surface. The ventral side of the female pointed upward and was therefore visible. Twelve scans of different mating complexes were performed with the Zeiss Xradia 520 Versa micro-CT scanner at Naturalis Biodiversity Center in order to adjust experimental conditions. Scan 14 was used to build the model of the female reproductive tract (Table S1). A total of 28 single fly specimen or mating complexes (Table S1) were scanned using Synchrotron Radiation X-ray Tomographic Microscopy (SRXTM) at the TOMCAT (X02DA) beamline of the Swiss Light Source, Paul Scherrer Institut, Switzerland. Of those, 26 scans could be used for data analysis (Table 1, Table S1). Absorption and phase contrast scanning modes were performed at 10, 20, and 40-fold magnification and 10 KeV, with 200 ms to 600 ms exposure time (Table S1). Projections were post-processed and rearranged into flat- and darkfield-corrected sinograms, and reconstruction was performed on a Linux PC farm, resulting in isotropic voxel dimensions of 0.65 µm, 0.33 µm and 0.16 µm.

Table 1 Number of scanned D. pachea copulation complexes and single females.

Virgin female	Virgin male	Copulation complexes, after copulation start (ACS)	Mated female 2 h ACS	Mated male 2 h ACS	
		2 min	8 min	15 min	20 min			
2	3	5	4	4	3	4	1	

Ejaculate quantification in female spermathecae

In the laboratory, D. pachea copulation lasts for about 30–40 min (Pitnick, 1993; Lang & Orgogozo, 2012; Rhebergen et al., 2016; Acurio et al., 2019; Lefèvre et al., 2021). To further examine when sperm transfer would take place during copulation, we interrupted copulation of a single couple by vigorous shaking at various time points after copulation start and examined the presence of ejaculate about 24 h later in the female sperm storage organs (Table S2). We prepared mating complexes as described above and monitored the mating progress by annotation of the courtship and copulation duration. The former was defined as the period from the first male “licking” courtship behavior (Speith, 1952) to the start of copulation. Copulation start was defined as the moment when the male mounted the female and copulation end was considered to be the moment when the male had dismounted, with genitalia and forelegs being fully detached from the female. Copulation was interrupted at 4 min, 8 min, 12 min, 18 min, and 24 min after copulation start by shaking the tube until the couple separated. The male was stored in 96% ethanol and the female was transferred into a food vial for 12–16 h at 25 ° C. The number of eggs laid were counted (Table S2) and the female paired spermathecae were prepared by opening the dorsal side of the abdomen and then separating them from the reproductive tract. The remainder of the female’s body was stored together with the corresponding male in 96% ethanol.

For imaging, spermathecae were transferred into a transparent dissection dish, filled with phosphate buffered saline (PBS, 137 mM NaCl, 2.7 mM KCl, 10 mM Na2HPO4, 1.8 mM KH2PO4, pH 7.4) and examined in lateral view with a VHX2000 (Keyence) microscope equipped with a 100-1000x VH-Z100W (Keyence) zoom objective at 500 fold magnification. For data analysis, file-names were replaced by a three-digit random number so that sperm presence was quantified blindly with respect to the annotated copulation duration. Ejaculate presence or absence in the spermathecae was annotated based on visual inspection according to previously published protocols (Jefferson, 1977; Lefèvre et al., 2021).

The uterus and/or the spermathecae of females of relevant synchrotron X-ray tomographic microscopy samples contained an apparent ejaculate mass that mainly formed spheres of about 5 µm. We counted these spheres to approximate sperm abundance and to compare the relative deposition of sperm in the left and right spermatheca (Table S1). For quantification, we used Imaris 9.9 software (https://imaris.oxinst.com/) to manually superimpose artificial spherical objects with identified sperm mass spheres, using the Imaris “spot” option.

Results

The male phallus is asymmetric

The phallus of D. pachea has a complex and asymmetric shape (Acurio et al., 2019) (Fig. 1). In dorsal/rostral view, the distal part (apex) is rather flat but pointed at the apical tip (Fig. 1B). It is strongly curved or folded at the base. The phallotrema is positioned dorso-apically on the right side of the phallus. In lateral view (Figs. 1A, 1C), a broadened ridge on the apical side and a slender ridge on the more distal side are observed. The ejaculatory duct is also located at the right side of the phallus and ends in the right-sided apical phallotrema (Figs. 1A, 1B). In contrast, the uterus of a virgin female appears to be bilaterally symmetrical (Fig. 2A). It appears to be S-shaped and forms a major sack with a dorsal and a ventral blind protrusion (Fig. 2B). The anterior part of the uterus forms a narrow tube that leads to the ovaries (ovaries are not included in the 3D model).

Figure 1 Asymmetric Drosophila pachea phallus morphology.

3D model of the male phallus (blue) and ejaculatory duct (light yellow). The terms left lateral, caudal, right lateral describe the orientation with respect to the male abdomen. The scale bar is 100 µm.

Figure 2 The female reproductive tract rearrangments during copulation.

3D models of virgin female D. pachea (A, B) and copulating couples (C–F). The female accessory glands and ovaries are not included into the model, (A) Virgin reproductive tract (pink) in caudal view, ovipositor (magenta) and spermathecae with spermathecae ducts (orange-gold). (B) Virgin uterus in lateral view (color codes as in A) and the seminal receptacle (yellow). (C) Copulating couple in lateral view, exterior cuticles of female and male (grey) semi-transparent and relative location of colored female reproductive tract and male external genitalia (green). (D–F) Copulation complex of C at higher magnification: female reproductive tract (pink), spermathecae and ducts (orange-gold), seminal receptacle (yellow) and male-phallus (blue), ejaculatory tract (light yellow) and genital arch (green), red lines indicate assigned compartment borders of the female reproductive tract into vagina, uterus and oviduct. (D) Dorsal view, (E) lateral view, (F) ventral view. The scale is 100 µm.

The reproductive tract rearranges during copulation

The uterus re-arranges and re-orients during copulation, similar to previous observations in females D. melanogaster Meigen (Adams & Wolfner, 2007; Mattei et al., 2015) (Fig. 2). Uterus and oviduct appear to be elongated along the antero-posterior axis of the female. The ventral blind protrusion, found in virgin females, is not detected and two constrictions are observed that allowed us to subdivide the female reproductive tract into three compartments (Fig. 2E, configuration at 20 min after copulation start): (1) the vagina, which is the most caudal part, (2) a balloon-shaped medial uterus, and (3) a broadened oviduct (Fig. 2D). The vagina appears to be elastic. It harbors and adopts the shape of the dorsal male phallus, which does not reach into the uterus but stretches the vagina in opposite dorsal-caudal direction. The male genital lobes (Fig. 2E) are positioned externally against the ventral female abdomen wall (not shown in the model). Together with the phallus they appear to be used to lever or clamp the male abdomen against the female abdomen (Figs. 2D, 2E). The caudal positon of the phallus tip with the male phallotrema is most distantly to the anterior parts of the female reproductive tract. Ejaculate needs to pass through the vagina and along the phallus towards the uterus in order to reach the female spermathecae (Fig. 2E). The vagina is dorsally pointed at its anterior end where it resembles the curved phallus base (Fig. 2E). The expanded medial uterus is separated from the anterior oviduct by a constriction, which also links the ducts to the spermathecae and the seminal receptacle at the dorsal and ventral side, respectively (Figs. 2D–2F). The oviduct forms a broad vesicle of the size of the uterus, whereas it is narrow in virgin females and in D. melanogaster (Adams & Wolfner, 2007) (Figs. 2B, 2E) (ovaries were not included in the model).

Different sperm morphologies in the female reproductive tract

Similar to previous findings (Pitnick, Markow & Spicer, 1999), no sperm cells were found in the seminal receptacle. By means of high-quality scans using SRXTM it was possible to visualize sperm mass within the female reproductive tract (Fig. 3, Table S1). The sperm cells appear to adopt different aggregation states in different parts of the uterus. They are spherically coiled or granular in the medial uterus (Fig. 3A), while they are filamentous when entering the spermathecae (Fig. 3B) and eventually re-adopt a spherical organization in the spermathecae (Fig. 3C). No distinction could be made between the sperm tails and heads and it is not clear if the identified spheres are filled or rather hollow “bubbles” that aggregate sperm tissue at the surface. Overall, it is not possible to deduce if the identified shapes correspond to single cells, but the number of spheres in the uterus and in the spermathecae appear to correlate with the total amount of sperm mass.

Figure 3 Sperm aggregates in the female reproductive tract.

(A) Female uterus harboring granular sperm aggregates, indicated by blue arrows. (B) Spermatheca with filamentous sperm (red arrows). Filamentous sperm that enters the spermatheca is indicated by yellow arrows. (C) Transverse section of a spermatheca, spherical organization of sperm mass is highlighted by red circles. The scale is 100 µm.

Sperm transfer can occur at 2 min after copulation start and is found in both spermathecae

We observed sperm presence in female spermathecae by dissections of females about 12–24 h after copulation. The spermathecae of virgin females (n = 9) were empty. Apparent sperm masses were identified after 4 min of copulation (2/12 trials) (Fig. 4). Overall, sperm transfer into the spermathecae occurred only in a minority of trials (13/64). Highest frequencies of ejaculate presence were observed after 8 min and 12 min (4/14 and 4/11 observations, respectively) of copulation. Unexpectedly, longer copulation duration of 18 min and 24 min yielded fewer cases of sperm presence in the spermathecae (2/12 observations and 1/15 observations, respectively). In all 13 trials with observable sperm transfer, both spermathecae revealed apparent sperm mass. In order to quantify different sperm levels in our synchrotron X-ray tomography scans, we examined the spherical organization of sperm mass in the uterus and in the spermathecae and counted the relative abundance of spheres in the left and right spermathecae (Table S1). Upon copulation, sperm was present in the female reproductive tract in the majority of scans (14/19 scans), but in different parts of the reproductive tract: in the uterus, spermathecae, spermathecal tubes, and in the broadened anterior part of the uterus (Tables S1). Sperm presence in the uterus was even detected in one couple at 2 min after copulation start (1/4 interpretable scans) and spermathecae contained sperm at least 8 min after copulation start (1/3 interpretable scans) (Table S1). Sperm sphere abundance varied between the left and the right spermatheca (Table S1) but no particular bias of sperm storage towards the left or the right spermatheca was observed among couples or at different time points after copulation start.

Figure 4 Ejaculate transfer occurs early during copulation.

Fraction of female spermathecae that revealed apparent sperm mass from copulations that were interrupted at different time points after copulation start.

Discussion

Genital asymmetry is not reflected in sperm allocation

Although we were unable to precisely quantify sperm, we observed no left or right-sided bias in the way sperm were deposited into the female reproductive tract. The distance between the male phallotrema and the female receptacles remains large at all stages of copulation, and it appears impossible for the asymmetric phallus to preferentially deposit sperm unilaterally in the female. Sperm allocation, may therefore not be controlled by the intermittent phallus morphology, but by female structures or by components of the male ejaculate. Copulatory and post-copulatory uterus rearrangements in D. melanogaster females are induced by male accessory gland proteins and are associated with sperm storage and fertilization (Adams & Wolfner, 2007; Mattei et al., 2015). Selective sperm storage has been associated with female muscular activity in female yellow dung flies Scathophaga stercoraria (L.), which was found to be sensitive to CO2 anesthetization and was proposed to rely on active female muscular movements (Hellriegel & Bernasconi, 2000). In a specific case, external and non-intermittent male genital asymmetry was found to indirectly stimulate postcopulatory sperm distribution in females of the fly Dryomyza anilis Fallén. During copulation, the male taps the female external genitalia after insemination with a pair of genital claspers (= surstyli: Rice et al., 2019), which can be symmetric or asymmetric. Male tapping affects sperm distribution in female sperm storage organs (Otronen & Siva-Jothy, 1991; Otronen, 1997) and males with asymmetric small claspers reveal higher fertilization success compared to males with large symmetric lobes since asymmetric male tapping probably affects sperm distribution in the spermathecae (Otronen, 1998). However, we know of no cases where the male phallus morphology influences sperm storage patterns in fly species. Phallus asymmetry in D. pachea may therefore rather ensure a relatively low basal or minimal efficiency of ejaculate transfer and stabilize the complex of male and female genitalia during copulation, but appears to be unrelated to postcopulatory sperm storage.

Phallus asymmetry is associated with a specific orientation inside the female uterus

We identified a curious orientation of the D. pachea male phallus inside the female uterus, which is strikingly different from previous findings in D. melanogaster. While in D. melanogaster and closely related species, the phallus points anteriorly towards the uterus and spermathecae (Yassin & Orgogozo, 2013; Mattei et al., 2015), it points to the caudal female body wall in D. pachea. The D. pachea phallus is curved at its base. It appears that the phallus functions on a mechanical basis to anchor the male abdomen on top of the female. Sperm is released from this most caudal position of the phallus and needs to pass the phallus base to reach the anterior uterus and female sperm storage organs. This is an intriguing observation since direct ejection of ejaculate into more anterior parts of the uterus would seem more appropriate for efficient gamete transfer. The copulation complex in D. pachea is shifted 6−8° to the right side with respect to the female antero-posterior midline (Rhebergen et al., 2016). Extrapolating this shift to the dorsal apex of the male phallus, the right-sided phallotrema is located in a medial position (Fig. 5). The dorsally pointed phallus structures stretch the dorsal vagina (like a covering sheet). The ventrally pointing asymmetric phallus spurs (Acurio et al., 2019) must be directed towards the caudal body wall and can potentially function as a grasping device to ensure an internal grip of the phallus. Alternatively, they may provide ventral flaps that avoid a close covering of the ventral vagina and therefore may stabilize a ridge in which ejaculate may be transported. Overall, the asymmetric phallus morphology may function to stretch the vagina in a specific configuration to enable efficient sperm release.

Figure 5 Asymmetric positioning of the male phallus.

Contour drawings of the dorsal apex of the D. pachea male phallus. (A) The male gonopore (filled black) is located at the right side of the phallus. (B) The twisted complex of female and male genitalia (Rhebergen et al., 2016) is expected to shift the male apex by 6–8° which should result in a medial positioning of the male gonopore with respect to the female antero-posterior midline (dashed line). The scale is 50 µm.

The uterus reshapes during sperm transfer

The uterus assumes a different shape during copulation compared to a virgin uterus, similar to previous observations in D. melanogaster (Adams & Wolfner, 2007; Mattei et al., 2015). However, broadening of the anterior part of the uterus was only observed in D. pachea. While it enlarges, the constriction at the junction to the spermathecal ducts remains narrow. This could potentially induce a vacuum and could either induce transport of the egg or perhaps result in a suction force to transport ejaculate towards the anterior part of the uterus. However, we have no data on the specific temporal dynamics of this re-arrangement. Future studies must analyze uterus shapes with progression of sperm and the eggs. Female D. pachea are rarely observed to lay eggs after a single copulation (Pitnick, 1993; Lefèvre et al., 2021), indicating that uterus reshaping might be related to other processes.

Low efficiency transfer of a complex ejaculate mass

Sperm transport into female spermathecae was observed only in a minority of mating trials. One possible explanation for this low rate is that our laboratory conditions may not provide optimal ecological parameters for this species and this may influence copulatory processes. Also, we use a laboratory stock that has been maintained for >20 years in captivity and which might reveal different sperm transfer frequencies with respect to wild D. pachea. Previous reports on D. pachea ejaculate transfer did not report that sperm is not always released from the male phallus during copulation (Pitnick, 1993; Pitnick & Markow, 1994). Nevertheless, our observations might indicate that ejaculation is, in fact, rare in D. pachea. Males of D. pachea were reported to transfer only 44 ± 6 sperm cells per copulation (Pitnick & Markow, 1994), which was found to be far below the female sperm storage capacity. Males produce giant sperm of >16 mm length (Pitnick & Markow, 1994) and the male investment into spermatogenesis was argued to be high compared with other Drosophila species. Thus, males might be selective with regard to the females to whom they would allocate their limited amount of gametes (Pitnick & Markow, 1994). Male gametes might even be so rare, that the females would require to mate multiple times to reproduce at all. Wild-caught D. pachea females were reported to contain sperm from at least 3–4 males and females are known to remate with various males (Pitnick & Markow, 1994). Alternatively, sperm might be discarded or absorbed by the female during or shortly after copulation or the female may prevent the male from ejaculating. Both might relate to female controlled cryptic mate-choice (Eberhard, 1996). However, in the majority of synchrotron X-ray tomography scans, sperm was present at least in some parts of the female reproductive tract. Our data indicates that sperm release from the phallus and sperm transport into the spermathecae might be two independent physiological processes. Again, the overall number of gametes in each ejaculate is low and females often remate. Thus, the relative loss of gametes per copulation is also low. Overall, the reproductive strategies for both sexes may imply a high level of promiscuity to compensate for a rather inefficient sperm transfer process during copulation.

However, when copulation does result in sperm transfer, it is released early during copulation, from at least 2 min after the start of copulation (the entire copulation lasts up to 40 min). Similar results were reported from remating experiments in D. melanogaster (Gilchrist & Partridge, 2000), where males required about 4–8 min to replace sperm from a previous mating. Intriguingly, longer copulation duration of D. pachea, of 18 min and 20 min yielded fewer cases of sperm transfer compared to shorter copulation durations of 8 min or 12 min. Previous analyses also revealed that copulation duration was negatively associated with sperm transfer rates in D. pachea (Lefèvre et al., 2021). Male dependent sperm allocation is not expected to show this pattern and was not observed in our X-ray computed tomography scans where the majority of scans revealed sperm presence in other parts of the female reproductive tract. Our observations indicate that sperm can be dumped by the female and this might occur if copulation takes too long, which can be a parameter of male performance during copulation (Eberhard, 1994). Alternatively, it could be argued that our data for the longer copulation durations are biased towards couples where the males continued copulating because they had not yet been able to transfer ejaculate. However, the investigated time-points were below the maximum copulation duration of 30–40 min seen in this species (Pitnick, 1993; Lang & Orgogozo, 2012; Rhebergen et al., 2016; Acurio et al., 2019; Lefèvre et al., 2021). We did not observe any couple that ended copulation before they were artificially separated.

Sperm was observed to be differentially organized in different organs. In the medial part of the uterus it had a granular shape while it was filamentous when entering the spermathecae. In D. melanogaster, sperm was reported to assume a torus shape inside the female uterus and such tori slowly rotate (Adams & Wolfner, 2007). It remains to be investigated at better resolution, if granular shaped sperm in D. pachea also approximates these tori conformations. What also needs further elaboration is whether these giant sperm are motile or are transported passively by other male ejaculate components or by female peristalsis. The aggregation changes might be related to the organization of the sperm tail, which may be non- or less motile. The filament perhaps undergoes dynamic aggregations or crosslinks with other sperm tails or with different parts of the tail. This could result in compaction and elongation of sperm mass and may explain the different aggregation states observed in our microCT and synchrotron X-ray tomographic scans.

Conclusions

The positioning of the asymmetric D. pachea male phallus was evaluated during copulation inside the female reproductive tract. The twisted copulation complex likely positions the right sided phallotrema into a medial location at the caudal end of the vagina. This particular positioning of the male phallotrema, the inflation of the oviduct during copulation and the low frequency of sperm transfer in different trials indicate a complex and inefficient sperm transport into female sperm storage organs during and after copulation. No evidence was obtained that the asymmetric male genitalia allow asymmetric sperm deposition inside the female reproductive tract. Future investigations must unravel if transfer of giant sperm might be non-autonomous in D. pachea, potentially mediated by re-arrangements of the female uterus and by different aggregation states of sperm during transport through the female reproductive tract.

Supplemental Information

Supplemental Information 1 microCT and synchrotron X-ray tomographic scans

Summary table of microCT and synchrotron X-ray tomographic scans and organ size measurements.

Click here for additional data file.

Supplemental Information 2 Copulation trials

Summary table of copulation trials to test for presence of sperm mass in the female spermathecae.

Click here for additional data file.

We would like to thank Maxi Richmond Polihronakis for discussions and for help with copulation complex preparation in D. pachea. We would also like to thank Kees Koops for his help and insight in starting and maintaining the fly population. We acknowledge the Paul Scherrer Institut, Villigen, Switzerland, for provision of synchrotron radiation beamtime at the TOMCAT (X02DA) beamline of the Swiss Light Source to MS and MR. We thank Dr. Federica Marone for supervision and assistance during the synchrotron experiments. We acknowledge the ImagoSeine core facility of the Institut Jacques Monod, member of IBiSA and France-BioImaging (ANR-10-INBS-04) infrastructures, in particular Vincent Contremoulins. We are grateful for comments from Patricia Brennan and two other reviewers. Lastly, we would like to thank Rob Langelaan and Bertie-Joan van Heuven for their help during the preparation and working with the microCT-scanner at Naturalis Biodiversity Center.

Additional Information and Declarations

Competing Interests

Author Contributions

Data Availability

The authors declare there are no competing interests.

Sanne van Gammeren conceived and designed the experiments, performed the experiments, analyzed the data, prepared figures and/or tables, authored or reviewed drafts of the article, and approved the final draft.

Michael Lang conceived and designed the experiments, performed the experiments, analyzed the data, prepared figures and/or tables, authored or reviewed drafts of the article, and approved the final draft.

Martin Rücklin analyzed the data, authored or reviewed drafts of the article, and approved the final draft.

Menno Schilthuizen conceived and designed the experiments, analyzed the data, authored or reviewed drafts of the article, and approved the final draft.

The following information was supplied regarding data availability:

The data, including images, 3D-scans and model, are available at Dryad: van Gammeren, Sanne; Lang, Michael; Rücklin, Martin; Schilthuizen, Menno (2022), No evidence for asymmetric sperm deposition in a species with asymmetric male genitalia, Dryad, Dataset, https://doi.org/10.5061/dryad.5x69p8d4z.

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
