# Peer review of "No evidence for asymmetric sperm deposition in a species with asymmetric male genitalia"

_PeerJ, doi:10.7717/peerj.14225_

## Round 0.1 · original submission · Major Revisions

The reviewers have responded positively to your paper and only one requires major revision.

Reviewer 1 ·

Basic reporting

No comment.

Experimental design

Some more details should be provided for i) the method of sperm count and ii) observation of egg laying.

Validity of the findings

The observations reported in this manuscript are interesting. However, I could not grasp a big picture view of why asymmetrical phallus has evolved in Drosophila pachea. If it functions only to compensate the asymmetrical mating posture of this species (Figure 5), why did this species abandon the ancestral combination of traits, that is, a symmetric phallus and symmetric external genital lobes, together with symmetric mating posture?

Additional comments

In this paper, van Gammeren and others examine the function of the laterally asymmetric phallus of Drosophila pachea using micro-CT techniques. Based on the observations, they conclude (1) that the phallus likely function to clamp the female body with the male external genitalia during copulation, and (2) that asymmetric phallus shape likely compensates the asymmetric mating posture of the species so that the male gonopore is placed almost on the mid line of the female body. The latter observation clearly rejects the hypothesis that the asymmetric phallus preferentially deliver sperm to one of multiple sperm storage organs of a female. The observations reported in this manuscript are interesting. However, I have some concerns and suggestions.

First, as written above, I could not grasp a big picture view of why asymmetrical phallus has evolved in Drosophila pachea. I suspect that, as in related species of D. pachea, their ancestors had symmetric lobes and thus showed symmetric mating posture. Because there was no need for compensation, the phallus would be also symmetric. The authors claim that asymmetry in the phallus shape evolved to compensate the asymmetric mating posture (caused by asymmetric lobes), which had been evolved for “some reason.” However, I think the opposite scenario is also possible: the asymmetric mating posture evolved to compensate the asymmetric phallus morphology, which had been evolved for “some reason.”

Considering the readership of PeerJ, Introduction and Discussion of the present form are somewhat too specific to the single studied species or dipteran insects. For example, though the authors write “we know of no cases where the male phallus morphology influences sperm storage patterns in fly species” (line 256-), what about other animal groups?

To my knowledge, this study is the first attempt to count the number of sperm cells in female organs using micro-CT techniques. The authors counted several hundreds of sperm cells based on f synchrotron X-ray tomographic sections. However, in the images shown in Figure 4, it is difficult to recognize sperm cells (at least for me). Much clearer images should be provided to show how sperm cells can be counted accurately.

Finally, many morphological terms used in the manuscript are specific for researchers of Drosophila fruit flies. I recommend incorporation of recent arrangements of genital nomenclature for Drosophila melanogaster (Rice et al. 2019). For example, instead of “genital arch”, “genital arch (epandrium)” will facilitate understanding by readers studying other insect groups.

minor points:
-line 146: 033 -> 0.33?

-line 160: “transferred into a food vial for AT LEAST 12 hours”: I could not understand this experimental design. Does this mean that females were allowed to lay eggs for different periods? If so, it is expected that females stayed longer in a food vial laid more eggs. Please explain.

-line 166: 2 and 4 of Na2HPO4 and KH2PO4 should be subscripts.

-line 174-: Is the spermatheca of D. pachea much longer than the sperm cells? If not, a single sperm cell can be counted twice or more. Please explain more details on how these data were analyzed. In addition, in Table S3, not only the sperm cells in the thin spermathecae, but also those in the uterus were counted.

-line 185: appearsto -> appears to

-line 190 (Results) and line 282 (Discussion): It is difficult to understand the change of uterus shape based on only Figure 2 and the description “as in D. melanogaster.” Please explain more details.

Figure 1: Please explain the orientation of images. For example, does the “left” side of “dorsal” image represent “right” side of the male body?

Supplementary tables are provided as MS Excel files. However, the values can be displayed differently in different environments. These should be provided as in PDF or in equivalent file formats.

·

Basic reporting

This article explores how asymmetric male genitalia function in Drosophila pachea, to determine is sperm deposition into the female reproductive organs is biased. The authors use micro-CT scans to determined that it is not, and the sperm ends up equally distributed in both sides of the female spermathecae when it is in fact transferred. They make other interesting discoveries about genital interactions between males and females during copulation. For example, the female oviduct widens significantly after copulation, and sperm is not transferred successfully in the majority of the copulations (75% or so).

The work is interesting and novel as the evolution of asymmetric genitalia is still little understood, and D. pachea has other interesting reproductive features for example giant sperm.

A potential consideration that is missing from the framework of the introduction is that the association between mating position and asymmetric genitalia may be female driven rather than male driven, for example if females who were mating from above (as suggested by Huber), evolve asymmetric genitalia to make sperm transfer more difficult, imposing selection on males to change their genitalia to adapt to the asymmetric female. This would force the males to always approach females from one side, and may reduce mating conflict if it gives females more control over copulation. This scenario has been recently proposed in harbor porpoises that have very asymmetric genitalia (Orbach et al 2020, full citation below). In this case, sperm transfer is not asymmetric in that there is only a single female reproductive channel, but barriers in the female vagina, may prevent the ejaculate from being deposited close to the cervix. In D. pachea, males always mate on the right side, and the females have an asymmetric twist of the female ovipositor, similar to what is described in porpoises.

For example in line 98-99, the authors write that “asymmetric male genitalia may have evolved to enhance the chances that sperm would be deposited in or near such preferential storage organs” but rather you could think about it from the female point of view: asymmetric female genitalia may make access to preferential sperm storage sites more difficult, forcing males to evolve asymmetric genitalia that try to regain access.

Experimental design

Good and standard for this taxa.

174-175: What does 25%, 50% and 75% of each spermatheca along the axis from the basis to the apical tip mean? I am confused. Are we talking about distance? Volume filled by sperm?

193-195: I am confused by the terminology. I am not an insect expert, but here you use the term vagina, uterus and oviduct, but I thought insects did not have either a vagina or a uterus? Also, examining Figure 2, and from the results, the phallus is only interacting with the vagina, which is a single channel leading to the uterus, where the entrances to the spermathecae are found… Seeing this then it would make sense that sperm may not end up in a single spermathecae because the ejaculation takes place in the single vagina.

200: spelling: position
242: Intromittent not intermittent

Validity of the findings

258: I am puzzled by this comment that asymmetric genitalia influences the efficiency of ejaculate transfer because sperm transfer seems to be so inefficient in this species (25% success).

304: But if males were this selective how are females getting sperm from 3-4 males in their oviduct? And why copulate at all if he is not going to transfer sperm?

310: I would argue Eberhard 1996 is a better citation for this than Eberhard 1985.

The inefficiency in sperm transfer suggests that females may have an effective system to prevent ejaculation, again thinking about this from a female, rather than a male perspective seems to make this conclusion less puzzling.

Additional comments

Orbach, D.N., Brennan, P.L.R., Hedrick, B.P., Keener, W., Weber, M.A. and S. L. Mesnick. 2020. Asymmetric and Spiraled Genitalia Coevolve with Unique Lateralized Mating Behavior. Science Reports 10, 3257. https://doi.org/10.1038/s41598-020-60287-w

Reviewer 3 ·

Basic reporting

no comment

Experimental design

no comment

Validity of the findings

no comment

Additional comments

This manuscript studies the changes that occur during mating with a male with asymmetrical male phallus. The study appears thorough and well presented and illustrated. Most of my comments are minor with some general edits added directly on the Word version of the manuscript. In addition, I have the following comments:
1. All references cited are listed in the Reference section. However, there is an inconsistency how the Journal titles are presented. Please check the journal style and instructions to authors.
2. No where in the paper is a reference to the source of terminology used. I suggest Rice et al. (2019) should be listed in the Materials and methods section.
3. The figure captions were not included on the Word version and I inserted Fig. 2 caption at the end of the ms with some spelling errors corrected and a comment.
4. The colour names were a bit confusing. The “orange” for the spermathecae appears to be more “gold” and the receptaculum is yellow rather than gold? At least to the reviewer.
5. The most significant comment concerns the use of the term “gonopore”. The true gonopore is at the opening of the ejaculatory duct and sperm pump and the “phallotrema” or secondary gonopore is at the apex of the phallus. The term that should be used throughout the ms is phallotrema. Reference: Kotrba (1993, Bonn. Zool. Mon. 33) and Sinclair (2000, Contributions to a Manual of Palaearctic Diptera, Vol. 1).
6. Taxonomic names should have the author listed when the name is first presented. I have added these to the ms.

Annotated reviews are not available for download in order to protect the identity of reviewers who chose to remain anonymous.

---

## Round 0.2 · Minor Revisions

Your manuscript has been much improved and now requires just some minor revision.

Reviewer 1 ·

Basic reporting

No comment.

Experimental design

No comment.

Validity of the findings

No comment.

Additional comments

I really appreciate the authors’ efforts to thoroughly address my questions to the previous version of the manuscript. This is a very interesting great work. Now I have only very minor comments as listed below.

<<Minor comments>>
Line 52: In Kamimura (2006), the importance of interlocking of female and male genitalia was not directly tested and illustrated.
The authors can cite Kamimura et al. (2019) instead.

Kamimura Y, Yang CCS, Lee CY. 2019. Fitness advantages of the biased use of paired laterally symmetrical penises in an insect. Journal of Evolutionary Biology 32: 844-855. DOI:10.1111/jeb.13486


Line 85: “genital arch (epandrium), nomenclature of terms according to (Rice et al., 2019)”
Since Rice et al. (2019) recommend to use “epandrium”, it would be better to write:
genital arch (= epandrium: Rice et al., 2019)


Line 209: “aedeagus” is used here instead of “phallus.” In D. melanogaster, the aedeagus is the main part of the phallus, which includes also some other lobes closely associated with the aedeagus (Rice et al. 2019). However, in D. pachea and the related species, I believe that these terms can be used interchangeably. Please consistently use “phallus,” and/or indicate “phallus (aedeagus)” around Line 87.


Line 280: “Selective sperm storage has been associated with female muscular activity.”
In what animals? In Drosophila melanogaster?


Line 286: As above, “genital claspers (= surstyli: Rice et al., 2019)” would be better.
Please note that not “surstili” but “surstyli.”


Line 345: “female controlled cryptic mate-choice” would be enough.

---

## Round 0.3 · accepted · Accept

Thank you for your submission and for making the required revisions.